# Insight into the B3Transcription Factor Superfamily and Expression Profiling of *B3* Genes in Axillary Buds after Topping in Tobacco (*Nicotiana tabacum* L.)

**DOI:** 10.3390/genes10020164

**Published:** 2019-02-20

**Authors:** Fei Xia, Tingting Sun, Shuangjuan Yang, Xiao Wang, Jiangtao Chao, Xiaoxu Li, Junhua Hu, Mengmeng Cui, Guanshan Liu, Dawei Wang, Yuhe Sun

**Affiliations:** 1Tobacco Research Institute, Chinese Academy of Agricultural Sciences, Qingdao 266101, China; xiafei1229@126.com (F.X.); wangxiao@zjtobacco.com (X.W.); chaojiangtao@caas.cn (J.C.); 82101171073@caas.cn (X.L.); hujh@wh.hbtobacco.cn (J.H.); cuimengmeng@caas.cn (M.C.); liuguanshan@caas.cn (G.L.); 2Key Laboratory for Tobacco Gene Resources, State Tobacco Monopoly Administration, Qingdao 266101, China; 3Graduate School of Chinese Academy of Agricultural Science, Beijing 100081, China; qing.dt@163.com; 4State Key Laboratory of Rice Biology, China National Rice Research Institute, Chinese Academy of Agricultural Sciences, Hangzhou 310006, China; 5Institute of Horticulture, Henan Academy of Agricultural Sciences, Zhengzhou 450002, China; sjyang_0614@163.com

**Keywords:** transcription factor, B3 superfamily, genome analysis, expression patterns, axillary buds

## Abstract

Members of the plant-specific B3 transcription factor superfamily play important roles in various growth and developmental processes in plants. Even though there are many valuable studies on *B3* genes in other species, little is known about the B3 superfamily in tobacco. We identified 114 B3 proteins from tobacco using comparative genome analysis. These proteins were classified into four subfamilies based on their phylogenetic relationships, and include the ARF, RAV, LAV, and REM subfamilies. The chromosomal locations, gene structures, conserved protein motifs, and sub-cellular localizations of the tobacco B3 proteins were analyzed. The patterns of exon-intron numbers and arrangement and the protein structures of the tobacco B3 proteins were in general agreement with their phylogenetic relationships. The expression patterns of 114 *B3* genes revealed that many *B3* genes show tissue-specific expression. The expression levels of *B3* genes in axillary buds after topping showed that the *REM* genes are mainly up-regulated in response to topping, while the *ARF* genes are down-regulated after topping.

## 1. Introduction

Transcription factors (TFs) are an important group of proteins that regulate gene expression at the transcriptional level by binding to specific DNA sequences. TFs participate in many biological processes as vital elements, functioning in a coordinated fashion to direct the cell life cycle, cell migration, and cellular organization during embryonic development and in response to signals from outside the cell. With the completion of genome sequencing in many plant species, genome-wide identification and analyses of genes encoding plant TFs is now straightforward [1]. In sixteen monocot and 38 eudicot species with genome sequences, the total number of TF genes identified ranged from 888 to 3714 [2]. Many TF families are present in plants as well as animals, bacteria, and yeast, including the homeodomain, MYB, bHLH, MADS, and bZIP families [1]. However, other classes of TFs are considered plant-specific, such as the AP2/EREBP, NAC, WRKY, GRAS, SBP and B3 families [1,3,4,5]. In our study, we chose to focus on the B3 transcription factor superfamily. 

All members of the B3 superfamily contain the plant-specific B3 domain, which has an average length of 100 amino acid residues. This domain was initially characterized in the *Viviparous1* (*VP1*) gene of maize [6] and the *VP1* orthologue *Abscisic acid-insensitive 3* (*ABI3*) from *Arabidopsis thaliana* [7] and possesses a sequence-specific DNA binding activity [5,8]. Several other domains, including APETALA2 (AP2), auxin response factor (ARF), auxin/indole-3-acetic acid (Aux/IAA), and zinc finger Cys- and Trp-containing domain (zf-CW), are also found in many multi-domain B3 proteins in addition to the B3 domain, and are considered to mediate protein-protein interaction and/or dimerization [9]. Based on the presence of these domains, the B3 superfamily can be classified into four distinct gene families, which include the LEAFY COTYLEDON2 [LEC2]-ABSCISIC ACID INSENSITIVE3 [ABI3]-VAL (LAV), RELATED TO ABI3 and VP1 (RAV), AUXIN RESPONSE FACTOR (ARF) and REPRODUCTIVE MERISTEM (REM) families [4].

The LAV family consists of the AFL and the VAL subfamily [4,10]. There are six *LAV* members in the Arabidopsis genome. Three well-studied genes, *ABI3*, *FUSCA3* (*FUS3*), and *LEAFY COTYLEDON2* (*LEC2*), belong to the *AFL* subgroup; these genes are known to control embryogenesis and the accumulation of storage reserves, and the genes show distinct temporal and spatial expression patterns during embryogenesis. In addition, the three AFL proteins can cooperate with each other and are influenced by mutual regulatory interactions during seed development and other developmental processes [10,11,12,13,14,15,16,17,18]. The three members of the VAL subgroup are VP1/ABI3-LIKE 1 (VAL1), VAL2 and VAL*3*, also known as HIGH-LEVEL EXPRESSION OF SUGAR-INDUCIBLE GENE 2 (HSI2), HSI2-LIKE 1 (HSL1) and HSI2-LIKE 2 (HSL2), respectively [19,20,21], and three other protein domains are present in addition to the B3 domain: a zf-CW domain, an ethylene response factor-associated amphiphilic repression (EAR) motif and a plant homeodomain-like (PHD-L) domain. Based on these domains, VAL proteins repress the seed maturation program and initiate germination and vegetative development by coordinating repression of the AFL network during seed germination rather than seed development [10,21,22,23,24,25,26,27].

The RAV family in Arabidopsis consists of 13 members, at least nine proteins of which have been demonstrated to play important roles in flower development. Six RAV members contain an additional AP2 domain in addition to the B3 domain [4,28,29,30,31]. The two domain-containing RAV proteins bind specifically to bipartite recognition sequence motifs [32]. TEM1 and TEM2 function in controlling floral transition in the photoperiod and gibberellins (GA) regulation pathways [33,34,35]. Other RAV proteins with AP2 domain (RAV1, RAV1L, TEM1/EDF1, and RAV2/TEM2) have negative regulatory effects on organ senescence and abiotic stresses [35,36,37,38], and RAV1 also regulates seed germination and early seedling development [39]. The *NGATHA* genes (*NGA1-NGA4*) are examples of the other seven *RAV* genes that only contain B3 domains and play redundant and essential roles in style specification and gynoecium development via the auxin signaling pathways [35,40,41,42,43]. The last three *RAV* family members (*ABS2/NGAL1*, *SOD7/NGAL2*, and *DPA4/NGAL3*), containing only the B3 domain, are involved in flower petal development and the regulation of seed size in plants [44,45]. *RAV* genes in other species have also been characterized and are suggested to play important regulatory roles in some general growth processes, such as the brassinosteroid biosynthetic and signaling pathways [46,47,48], bud outgrowth [49,50], lateral organ development [51,52], and photoperiod [50,53,54]. In addition, expression of these genes responds to pathogen infections [55,56] and abiotic stresses [35,57]. 

The ARF family is the best-studied subfamily of B3 superfamily, and ARF proteins regulate the expression of auxin-responsive genes through specific binding to auxin-responsive elements (AuxREs; TGTCTC) located upstream of auxin-responsive genes [58,59,60,61,62,63]. Since the first ARF protein was identified from Arabidopsis over 20 years ago [62], ARFs from more than 30 dicots and monocots have been widely characterized based on the sequence similarity to AtARF1 and genome-wide analysis, with the numbers ranging from eleven to 56 [58,60,64,65,66,67,68,69]. The ARF subfamily contains both transcriptional activators and repressors, and both of them can act cooperatively as dimers or oligomers [4,9,58,63,70,71]. Genetic and molecular studies in Arabidopsis and other plant species have suggested that ARFs are key factors in the regulatory networks that control plant growth by regulating developmental and physiological processes that include embryo patterning [72,73], organ polarity [74,75], leaf senescence [76,77], lateral root formation [78,79,80], floral morphogenesis [77,81,82], fruit development [83,84,85,86,87], and abiotic stress responses [88,89,90,91]. ARFs can also integrate multiple signaling involving hormones such as ethylene [92,93], brassinosteroid [80,94,95,96], and GA [97,98].

The REM family represents the most diverse subfamily in the B3 superfamily. Relatively few members have been functionally characterized, which is inconsistent with the large number of REM proteins. All REMs contain one to seven B3 domains, which are often present in more than two copies. The B3 domains present in the REM proteins show variation with respect to amino acid sequence and length [30,99]. Moreover, *REM* genes are phylogenetically divergent and extensively duplicated, and are mostly located in clusters in the Arabidopsis genome [4,30,100]. The first *REM* gene isolated, *BoREM1*, is expressed specifically in reproductive meristems in cauliflower, and was proposed to be involved in the determining the identity of the floral meristem [101]. *AtREM1*, the Arabidopsis ortholog of *BoREM1*, is preferentially transcribed in reproductive meristems, and functions in the floral organ development [102]. *VRN1* functions in maintenance of the vernalization response and is proposed to be involved in the epigenetic repression of FLOWERING LOCUS C (FLC) in a non-sequence-specific DNA binding manner [99,103,104,105,106]. Another *REM* gene, *VERDANDI* (*VDD*), plays a role in female gametophyte development in Arabidopsis, as a direct target of the MADS-domain ovule identity complex that includes SEEDSTICK (STK), SEPALLATA3 (SEP3) and SHATTERPROOF1/2 (SHP1/2) [107,108,109]. Six *REM* genes are differentially expressed in the SHP2 expression domain [110]. Moreover, eight *REM* genes were demonstrated to be targets of the floral homeotic MADS transcription factor AGAMOUS [111].

Tobacco is cultivated as an economic crop, a potential bioenergy crop and is an important research model plant for studying fundamental biological processes and plant disease susceptibility [112,113,114,115]. During growth, the shoot architecture of tobacco is controlled by apical dominance and development process. After removal of the apical dominance by topping (removing the inflorescence just before flowering), the axillary buds rapidly grow into branches, which can severely affect the biologic and economic yields of tobacco. At present, few genes involved in the development of axillary buds have been characterized in tobacco. It is well known that B3 TFs have multiple functions in various pathways that regulate branch development by controlling hormone response genes, sugar response genes, and meristem developmental genes. However, the *B3* gene families remain relatively poorly characterized in tobacco. Therefore, it is a necessary first step to analyze the *B3* gene families at a genome-wide level. Fortunately, the genomes of four species of Nicotiana, including several cultivars of the allotetraploid tobacco have been sequenced [115,116,117,118]. In addition, mutant libraries of tobacco based on both activation-tagging and chemical mutagenesis have been constructed and characterized [119,120]. These resources provide an opportunity to enhance our understanding of the function of *B3* gene families in branching development in tobacco compared with those of Arabidopsis.

In this work, we identified 114 *B3* genes in the tobacco genome using comparative genome analysis. The evolutionary relationships and gene structures were investigated to gain insight into the role of *B3* genes in regulating the processes of growth and development in tobacco. The expression patterns of 114 *B3* genes analyzed using RNA-seq data from common tobacco revealed that many *B3* genes show tissue-specific expression. The expression levels of *B3* genes in axillary buds were assayed by Quantitative Reverse Transcription Polymerase chain reaction (qRT-PCR) after topping treatment, which showed that the expression of *REM* genes mainly responded positively to topping; on the contrary, *ARF* genes were down-regulated after topping.

## 2. Materials and Methods 

### 2.1. Identification and Classification of *B3TF* Family Members in Tobacco

Whole genome protein sequences from tobacco and Arabidopsis were downloaded from http://solgenomics.net/organism/Nicotiana_tabacum/genome (SNG database) and http://www.arabidopsis.org/. The following strategy was used to analyze each *B3* gene from the genome of tobacco. The Hidden Markov Model (HMM) profile of the B3 domain (PF02362) was obtained from the Pfam database (http://pfam.sanger.ac.uk). The conserved B3 domain was used to search for B3 protein sequences using HMMER software with default parameters. B3 domain in the proteins obtained was predicted and identified again, and the proteins with no B3 domains were eliminated. The B3 proteins in tobacco and Arabidopsis were compared to identify orthologs by searching a local database that was built using Arabidopsis B3 proteins, and the E-value was set at 1e-10. BLASTP and TBLASTN with default parameters were also performed to identify further the NtB3 proteins using AtB3 protein sequences from tobacco protein and mRNA sequences, respectively. After removing the redundant sequences, all of the non-redundant and high-confidence proteins were assigned as tobacco B3 family members. The isoelectric points and protein molecular weights were predicted with the ExPASy Proteomics analysis tools (http://expasy.org). The sub-cellular localizations were predicted using the Protein Subcellular Localization Prediction Tool (https://www.genscript.com/psort.html).When the subcellular locations of TFs were not predicted in the nuclear, the ortholog B3 proteins of Arabidopsis were localized according to the data of TAIR and NCBI.

### 2.2. Multiple Sequence Alignments and Phylogenetic Analysis

Multiple sequence alignments of the predicted total B3 proteins in tobacco were conducted using ClustalW with the default pairwise and multiple alignment parameters. The protein sequences of each of the LAV, RAV, and ARF subfamilies from both tobacco and Arabidopsis were aligned using ClustalW with default parameters. In addition, multiple sequence alignments of the REM subfamilies from both tobacco and Arabidopsis were performed using Muscle with default parameters, as the alignments of the REMs based on ClustalW failed to be conducted because of the complexity of sequence similarity. Phylogenetic trees were generated using the ML (Maximum likelihood) method as implemented in MEGA 6.0 with 1000 bootstrap replicates based on the JTT matrix-based model [121,122].

### 2.3. Chromosomal Location

The chromosomal locations of the *B3* genes were retrieved from the SGN database. The physical positions of the *B3* genes in identified tobacco were mapped to the chromosomes with MapDraw V2.1 software [123]. 

### 2.4. Gene Structure and Conserved Motif Analysis

The structures of the *B3* genes were analyzed and illustrated using the Gene Structure Display Server (GSDS) tool (http://gsds.cbi.pku.edu.cn; [124]). The distributions of the conserved motifs in the tobacco B3 proteins were predicted with MEME 5.0.2 (Multiple Expectation Maximization for Motif Elicitation, http://meme-suite.org/) using the following parameters: the optimum motif width was set from 6 to 200, and the maximum number of motifs to identify was set to 15 motifs. 

### 2.5. Expression Pattern Analysis of Tobacco *B3* Genes

High-throughput RNA-seq data of *Nicotiana tabacum* (*N. tabacum*) was downloaded from https://www.ncbi.nlm.nih.gov/sra. The RNA-seq data for *B3* genes in 11 tobacco tissues were extracted and used to analyze the expression of *B3* genes in different tissues by the FPKM (fragments per kilobase per million reads mapped) method. To investigate the tissue-specific expression patterns of *B3* genes in tobacco, we analyzed and extracted the data of the *B3* genes separately from each tissue included in this study with a consistent method using Tophat and Cufflinks in the R statistical computing environment. To visualize the expression pattern of *B3* genes in different tissues, the treated expression data were standardized and analyzed using Cluster software, and then processed into a CDT profile, which could then be used to generate the Heat Map by Treeview V1.1.6 software.

### 2.6. Plant Growth, Topping Treatment, and Quantitative Real-Time PCR Analysis

The *N. tabacum* flue-cured cultivar K326 was used to study *B3* gene expression in this study. The plants were grown in the greenhouse, and two groups of rapidly growing wild-type plants were used in this research. One group was allowed to grow normally without topping while the other group was topped, and the axillary buds from the upper and lower parts of the tobacco plants in the two groups were then collected for RNA extraction to analyze the expression of tobacco *B3* genes. Total RNA was extracted using the Plant RNA Extraction Kit (TaKaRa, Dalian, China) with three biological replicates according to the manufacturer’s instructions. We used qRT-PCR to determine the relative mRNA levels of the genes, which were calculated using the 2^−ΔΔCt^ method. The gene-specific primers given in Appendix A based on the 114 non-coding regions of the *B3* genes were designed for qRT-PCR specifically to avoid non-specific amplification.

## 3. Results

### 3.1. Identification of *B3* Genes in the Tobacco Genome

In this study, 114 B3 tobacco genes were identified through extensive searches for B3-type domains and the annotation information for *B3* genes in the SGN database. The characteristic features of *B3* genes are listed in Table 1 and Appendix A. The lengths of proteins encoded by the *B3* genes varied from 99 to 1106 amino acids, with predicted protein masses ranging from 11,220.96 kD to 122,010.84 kD. The theoretical isoelectric points (pI) ranged from 4.5 to 9.8. The results of sub-cellular localization prediction suggested that 89 B3 proteins are localized to the nuclear, five are localized to the chloroplast, one protein is localized to mitochondrion, and the remaining 21 proteins are localized in the cytoplasm (Table 1, Appendix A). However, the Arabidopsis ortholog proteins of five tobacco ARFs localized to the cytoplasmic and chloroplast, ten REMs localized to the cytoplasmic, and two REMs localized to the chloroplast were all localized to the nuclear according to the data of TAIR and NCBI. Arabidopsis ortholog proteins of six tobacco REMs localized to the cytoplasmic were all localized to the chloroplast (Appendix A).The results indicated that the predictions of subcellular locations of ortholog B3 proteins are not completely identical in different species.

To survey the size characteristics of B3 proteins in plants, the findings of some previous studies on *B3* genes in several representative species, including the dicot species Arabidopsis (*Brassicaceae*) and tomato (*Solanaceae*), and the monocot rice, are summarized in Table 2. The total number of predicted *B3* genes in tobacco was found to be similar to the numbers in Arabidopsis (118), rice (91), and tomato (97). However, the number of *ARFs* found in tobacco was double that found in tomato. Moreover, the numbers of *ARF* subfamilies in the parental species of allotetraploid tobacco were both twenty-three, nearly half that of *N. tabacum*. The numbers of *RAV* and *LAV* subfamilies in the parental species were all four, compared with five members in allotetraploid tobacco. The numbers of *REM* subfamilies in the parental species were forty-four and fifteen, respectively, of which total number is equal to that found in allotetraploid tobacco. The statistics of the number of *B3* genes in four species showed that the distribution of *B3* genes into the four subfamilies in different species is similar, especially when comparing tobacco with tomato, regardless of the total number of *B3* genes or the number of genes in each subfamily.

### 3.2. Phylogenetic Relationship Analysis of *B3* Proteins in Tobacco

To investigate the phylogenetic relationships among the tobacco *B3* genes, a local Arabidopsis B3-family protein database was searched with the 114 tobacco B3 protein sequences to identify the orthologous genes in the tobacco B3 family. Based on the phylogenetic classification of the B3 superfamily in Arabidopsis, the 114 tobacco B3 proteins were classified into four groups, as in Arabidopsis (Figure 1, Appendix A). A phylogenetic analysis was then performed based on the multiple sequence alignment of all tobacco B3 proteins. The phylogenetic tree constructed using the Maximum Likelihood method clustered the B3 proteins into four large clades (Figure 1). There are five proteins classified in the LAV subfamily, five proteins in the RAV subfamily, 45 proteins in the ARF subfamily, and the remaining 59 proteins were classified into the REM subfamily, which is considerably larger than the others (Figure 1, Appendix A). The members of the REM subfamily in Arabidopsis is inconsistent with different studies that used different classification standards [4,9,30], showing REM proteins to be a very complicated group, and that the classification is no longer apparent. In our study, we did not further classify the REM family into smaller groups. In summary, similar to surveys on B3 superfamily proteins in other plant species, the REM subfamily contains the largest number of proteins of the four subfamilies in the B3 superfamily, and the ARF subfamily also contains a large number of proteins. Both the LAV and RAV subfamilies are small, with just five members each, considerably fewer than in either the ARF or REM subfamilies. Distribution of the 114 tobacco B3 proteins in four subfamilies is consistent with the classification of B3 proteins in three other species, especially in tomato, which, like tobacco, belongs to the botanical family Solanaceae (Table 2). These results suggested that the tobacco protein sequences and their classification are reliable, and that they could therefore be used in subsequent analyses.

To gain further insight into the evolutionary relationships among the 114 tobacco B3 proteins, ML bootstrap consensus trees for each B3 protein subfamily in tobacco and Arabidopsis were generated using Clustal (Appendix A) and Muscle (Appendix A). These classifications showed that NtLAV3 and NtLAV4 are in a clade with VAL2 of Arabidopsis, and NtLAV1 and NtLAV5 are more closely related to VAL3 in *A. thaliana* (Appendix A). In the RAV subfamily, NtRAV1, 2, and 5 group in a large clade with the Arabidopsis NGA proteins, and NtRAV3, and NtRAV4 are in another large clade with three AtNGAL proteins (Appendix A). Overall, the tobacco ARFs show closer evolutionary relationships to the ARFs of Arabidopsis compared to the evolutionary relationships of the other three subfamilies between tobacco and Arabidopsis (Appendix A). There are 13 tobacco proteins that have >90% sequence homology with Arabidopsis proteins (Appendix A). However, the evolutionary relationships of the REMs between tobacco and Arabidopsis are very low (Appendix A).

### 3.3. Chromosomal Location of *B3* Genes in the Tobacco Genome

The chromosomal distribution results revealed that 54 of the *B3* genes were unevenly distributed on 18 tobacco chromosomes, 12 genes were anchored to unattributed scaffolds that consist of only scaffold length and ID, and 17 genes had no chromosomal location information in the tobacco genome (Figure 2, Appendix A). No *B3* genes were mapped to six tobacco chromosomes, including chromosomes 6, 7, 8, 16, 20, and 21. Tobacco chromosome 17 has 10 *B3* genes, which is the most of any chromosome. We also found that chromosome 17 is the longest among all of the 24 tobacco chromosomes, and that the *B3* genes located on this chromosome mainly consist of *REM* and *ARF* subfamily genes. Chromosomes 12, 15, and 19 also have a large number of *B3* genes, containing 5, 6, and 5 genes, respectively. Five chromosomes (1, 4, 5, 11, and 24) have only a single gene each, which is the lowest number (Figure 2, Appendix A). Thirty-one of the *B3* genes were found to have 1-3 homologous genes (Appendix A). Both *NtREM22* and *NtREM41* have three identical homologous genes, including *NtREM53*, *NtREM56*, and *NtREM57*. *NtREM24*, *NtREM25*, and *NtREM40* have two homologous genes (Appendix A).

### 3.4. Gene Structure and Conserved Motif Analysis of *B3* Superfamily Proteins

To understand the structural diversity of the tobacco *B3* genes, we analyzed the exon/intron arrangements in the coding regions. As a whole, the exon/intron numbers were high, and they varied from 2 to 15 in the *B3* superfamily genes. The *ARF* subfamily members are predicted to have 2–15 exons, with an average of 10.64. While 11 of the *ARF* genes have only 4–5 exons, most of 45 tobacco *ARF* genes have more than 10 exons. The *NtARF22* in individual clade have the minimum number of exons. Genes located on the same branches of the phylogenetic tree share similar gene structures and exon/intron numbers (Figure 3). The five members of the *RAV* subfamily have 2–3 exons each, and the gene structures are also similar within the clade. All tobacco *LAV* genes have more than 10 exons except for *NtLAV4*, which contains six exons, and the exon/intron organizations of the *LAV* genes are consistent. However, the *REM* subfamily members have diverse gene models and exon/intron numbers. The genes in the *REM* subfamily have 2–11 predicted exons, with an average of 6.6. In the *REM* subfamily, the exon numbers may differ between some neighboring genes in the phylogenetic tree. In the B3 protein family, the *REM* subfamily is the most numerous and divergent, not only in tobacco, but in all species in which these genes have been studied (Figure 3).

The functional conserved domains in the tobacco B3 proteins were compared and analyzed (Appendix A). In ARF subfamily, all members have a B3 domain and a middle domain, except for NtARF19 and NtARF40 (Appendix A). The NtARF19 and NtARF40 proteins are short and have only a B3 domain without the auxin response factor and AUX_IAA domains (Appendix A). The B3 domain and middle domain in most of the ARF proteins are located close to the N termini (Appendix A). Twenty-two members of the ARF subfamily contained all three motifs, and these proteins were mainly distributed in three large clades (Appendix A). In general, proteins that show close evolutionary relationships were always similar with respect to the types and quantities of motifs they contained. The five RAV proteins have only a B3 domain (Appendix A). Three members of the LAV subfamily possess both a B3 domain and a zf-CW (CW-type zinc-finger) domain, while the remaining two members only contain a B3 domain (Appendix A). The REM proteins each contain 1–5 B3 domains, and they are evenly distributed in the REM proteins that contain between 3 and 5 B3 domains. In the REM proteins that contain two B3 domains, they are mainly found in the N-terminal and C-terminal regions (Appendix A). The B3 domains are located in the N-termini and middle regions in most of the REM proteins that have a single B3 domain (Appendix A). The similar distributions and numbers of B3 domains in REM proteins were congruent with the branch distances in the evolutionary tree between two REM proteins. The length variations in several of the domains in the B3 superfamily may affect the exact core structure of the proteins [9]. In tobacco, the length of the B3 domain varies from 41 to 106 amino acids, with an average of 90.9, and the ARF and AUX_IAA domains also show similar length variations that range from 30 to 83 and 38 to 115 amino acids, respectively. The length of the zf-CW domain in the three LAV proteins was very consistent, at 42 amino acids in all three proteins (Appendix A).

To further explore the structure of the tobacco B3 proteins, the conserved motifs of the B3 proteins were analyzed based on the phylogenetic relationships. Ten conserved motifs, which we named 1 to 10, were identified using the MEME tool (Figure 4 and Figure 5, Appendix A). Proteins in the same clades tended to have similar motif types and arrangements (Figure 4), which supports the results of the gene structure analysis and the phylogenetic analysis. In addition, we conducted analyses of the conserved B3, ARF, and Aux/IAA domains (Appendix A), and found that Motifs 1,2,6 are fragments on the B3 domain, motif 10 is the ARF domain, and Motif 8 is the Aux/IAA domain (Figure 5, Appendix A).

### 3.5. Tissue-Specific Expression Profiling of the *B3* Genes in Tobacco

The expression patterns of the *B3* genes at different developmental stages were investigated by analyzing the RNA-sequencing data in 12 different tobacco tissues that represent the entire tobacco growth cycle (Figure 6 and Figure 7). In tobacco seedlings, the highly expressed genes in the shoot apex and shoot were mainly in the *ARF* subfamily. The *ARF* genes *NtARF35*, *NtARF12*, *NtARF18*, *NtARF25*, *NtARF4*, *NtARF31*, *NtARF6*, and *NtARF45* are expressed at high levels in the shoot apex, while *NtARF9*, *NtARF38*, *NtARF28*, *NtARF32*, *NtARF35*, and *NtRAV1* show high expression levels in the seedling shoot (Figure 6). In seedling roots, in addition to the *ARF* genes *NtARF14*, *NtARF23*, and *NtARF2*, several *REM* genes are also expressed at high levels, such as *NtREM37* and *NtREM38* (Figure 6). A few genes in the shoot are expressed at similar levels in the shoot apex; however, these genes exhibited the opposite expression pattern in root tissue compared with the other two tissues. An example is *NtREM37*, which suggests that expression of these genes is tissue-specific during the seedling stage in tobacco. In summary, most of the genes expressed in the shoot apex are tissue specific. There are 10 *ARF* and 13 *REM* genes show shoot-specific expression, and both the *RAV* and *LAV* families have one member that is specifically expressed in the shoot. Genes that show root-specific expression are found in both the *ARF* and *REM* families; notably, the *LAV* family contains two genes that are expressed specifically in the root (Figure 6). In adult plants, *NtARF12*, *NtARF13*, *NtARF31*, *NtARF40*, *NtARF44*, *NtREM26*, *NtREM3*, *NtREM43*, and *NtREM19* are expressed at very low levels in all tissues except flowers, including young flowers, mature flowers, and senescent flowers (Figure 7). The *NtARF12*, *NtARF13*, *NtARF31*, and *NtREM26* genes show their highest expression levels in young flowers, and are also expressed at high levels in senescent flowers. Similarly, *NtREM3*, *NtREM43*, *NtREM19*, *NtARF40*, *NtARF2*, and *NtARF25* have their highest expression levels in mature flowers, and also show high levels of expression in senescent flowers. *NtREM19* and *NtARF4* are expressed at high levels in mature flowers and senescent flowers. As shown in Figure 7, over the blooming cycle of the flower, the expression levels of several *ARF* genes including *NtARF23*, *NtARF45*, *NtARF45* and *NtARF5* decreased gradually, while the relative expression of some *REM* genes, including *NtREM15*, *NtREM41*, *NtREM27*, and *NtREM18*, increased. We also found that for genes with high expression levels in young flowers, expression was much higher than in mature and senescent flowers. In the dry seed capsule, several genes from the *REM* subfamily show very high expression levels; these include *NtREM33*, *NtREM4*, *NtREM55*, *NtREM10*, *NtREM36*, and *NtREM37*. In both the mature and senescent leaf, *B3* genes have similar expression profiles. During leaf senescence, the expression levels of *NtARF21*, *NtARF11*, *NtARF16*, *NtARF28 NtREM36*, and *NtREM37* declined. *NtARF22* and *NtRAV2* were found to be expressed in mature leaves, and *NtARF43* showed a high level of expression in senescent leaves. In general, *NtARF12*, *NtARF13*, *NtARF31*, *NtARF40*, *NtREM26*, *NtREM3*, *NtREM43*, and *NtREM19* show flower-specific expression, and the expression of *NtREM4*, *NtREM9*, *NtREM10*, *NtREM55*, and *NtLAV3* is specific to the dry capsule. *NtRAV2* is specifically expressed in the mature leaf, and the expression of *NtREM30* and *NtREM31* is specific to the root (Figure 7).

### 3.6. Expression Profiles of *B3* Genes in Tobacco Axillary Buds in Response to Topping

Topping is widely practiced in tobacco cultivation, and tobacco axillary buds grow rapidly after topping. Therefore, we quantified the expression levels of all newly identified tobacco *B3* genes in the upper and lower parts of the tobacco axillary buds after topping using qRT-PCR (Figure 8 and Figure 9, Appendix A). No SYBR Green signals were detected for four genes (*NtARF19*, *NtARF42*, *NtREM14*, and *NtREM39*), while normal signals were detected for the internal control gene. Comparing the results of the *B3* gene expression levels between HB1U (upper axillary buds before topping) and HB2U (upper axillary buds after topping) showed that the expression levels of *NtREM8*, *NtREM30*, *NtREM33*, *NtREM35*, *NtREM36*, *NtREM37*, *NtREM47*, and *NtREM48* increased, while the expression levels of 13 genes including *NtREM16*, *NtREM25*, *NtREM27*, *NtREM54*, *NtREM57*, *NtARF1*, *NtARF22*, *NtARF31*, *NtARF35*, *NtARF40*, *NtARF41*, *NtLAV5*, and *NtRAV1* decreased (Figure 9). We also compared the expression levels of *B3* genes between HB1D (lower axillary buds before topping) and HB2D (lower axillary buds after topping), and found that in this group, the relative expression of eight genes (*NtARF32*, *NtARF39*, *NtREM20*, *NtREM26*, *NtREM40*, *NtREM55*, *NtREM56*, and *NtREM59*) increased, while expression of six genes (*NtRAV2*, *NtRAV5*, *NtREM31*, *NtARF28*, *NtARF30*, and *NtARF33*) decreased (Figure 9). It is worth noting that there are six genes in which expression changed significantly in these two types of comparisons (Figure 9). Among of them, *NtREM33* and *NtREM48* were up-regulated, while *NtARF31* was down-regulated, in both the upper and lower axillary buds after topping. Expression of the remaining four genes was up-regulated in the upper axillary buds but was down-regulated in the lower axillary buds. The relative expression of many *B3* genes changed significantly in the axillary buds after topping; this was especially true for *NtREM8*, *NtREM30*, *NtREM32*, *NtREM36*, *NtREM37*, and *NtREM56*, where expression increased by 23-, 9-, 8-, 8-, and 8-fold, respectively, compared to expression before topping. Also, the relative changes in expression after topping were 7-fold for *NtRAV1* and 14-fold for *NtARF30* compared to before topping (Figure 9). From these results, we found that many genes showed significant changes in expression after topping, and the responses of these genes to topping were not identical in the upper and lower axillary buds of tobacco (Figure 8 and Figure 9).

## 4. Discussion

The plant-specific B3 superfamily, which was defined because the B3 proteins contain at least one B3 domain, is one of the largest families of transcription factors, and its members play important roles in phytohormone, sugar, and other signaling pathways involved in plant growth and development [12,13,17,18,33,125]. Although there have been several previous studies of the B3 family in plants, our study is the first to investigate and characterize the B3 family genes in tobacco. In this study, we identified 114 *B3* genes through a comprehensive search for the B3 protein domain, and a phylogenetic analysis grouped the proteins into four subfamilies that reflected the grouping of the Arabidopsis B3 proteins [4], and included the ARF, LAV, RAV, and REM subfamilies. Although tobacco is an allotetraploid plant (2n = 48), and the number of chromosomes in tobacco is twice the number found in tomato and rice, the total number of *B3* genes in tobacco is not much more than that in tomato and rice, and fewer than that in Arabidopsis. However, the number of ARF in parental species of the allotetraploid tobacco was both the same with that of tomato and nearly half to that of *N. tabacum*. The numbers of RAV and LAV subfamilies in parental species were nearly the same numbers as those in tomato and the allotetraploid tobacco. The numbers belonging to the REM superfamily is variable in different species, and relatively, very few members of the REM subfamily have been functionally characterized. Therefore, based on the distribution and numbers of the ARF, LAV, and RAV subfamilies in different species, and the close genetic relationship between the parental species of the allotetraploid tobacco, we can speculate that the genome of allotetraploid tobacco has not yet been perfectly assembled and annotated in the SNG database.

The B3 proteins mainly contain five types of functional domains: B3, AP2, zf-CW, ARF, and Aux/IAA. The B3 domain possesses sequence-specific DNA binding activity and tends to recognize the CACCTG sequence in *AtRAV1* [32]. In rice, experimental evidence has shown that the B3 domain contributes to nuclear localization of ARF proteins in addition to the DNA binding activity [126]. In our research, we found that the B3 domain is mainly located in the N-terminal region of B3 proteins, except for proteins in the REM subfamily. It is known that some localization-related signal sequences are often located at the N-termini of proteins. Most of the REM proteins contain more than one B3 domain, and show variation in the sequence and length of the B3 domains, which we found to be the case with tobacco REM proteins (Appendix A). These multiple B3 domains in proteins could be derived from one or more B3 domain duplication events. During evolution, domain duplications can generate new genes that enable the plant to respond to external stimuli, and also enrich the functions of proteins in a gene family, thereby increasing functional diversity [9]. However, the functional characteristics of these proteins with domain duplications are not easy to determine, because many of the REM proteins are functionally redundant. The functions of only two *REM* genes have been well studied to date. In addition, the crystal structures of RAV1 and AT1G16640.1 from Arabidopsis showed that an open β-barrel consisting of seven β-strands and two α-helices between the β-strands constitutes the primary structure of the B3 domain [4]. This tertiary structural model was verified in different members of the B3 family by modeling, and the tertiary structure of the B3 domains is similar in the different subfamilies of the B3 superfamily, even though the protein sequence similarities are low [4,31], which shows that the structure of the B3 domains is well conserved. The zf-CW domain is present in most of the VAL proteins in the LAV subfamily, and possesses DNA-binding activity. In tobacco, we identified five LAV proteins, of which three members, NtLAV2, 3, and 5, have a zf-CW domain. Interestingly, the evolutionary relationships between these three proteins are not very close, but NtLAV1 and NtLAV3 share genetic relationships of >99% with NtLAV5 and NtLAV4, respectively; two proteins without zf-CW domains. We speculate that loss of the zf-CW domain in NtLAV5 and NtLAV4 occurred during evolution. The AP2 domain, which plays an important role in the development of the meristem, floral organs, and seed coat in Arabidopsis, is another DNA binding domain found in some RAV proteins in addition to the B3 domain. RAV proteins mainly include two types, one type has an AP2 domain, and the other does not. In our study, the five tobacco RAV members do not contain AP2 domains. Based on the evolutionary relationships between the RAV proteins of Arabidopsis and tobacco, the five tobacco RAV proteins are closer to NGA and NGAL in Arabidopsis, which do not have AP2 domains. In Arabidopsis, NGALs are involved in flower and seed development, while the NGA proteins are mainly involved in flower and leaf development and regulation of leaf morphogenesis [44,45]. Therefore, the *RAV* genes of tobacco may have significant roles in flower, seed, and leaf development. The middle domain and Aux/IAA domain are both found in ARF proteins. In tobacco, the middle domain is located at the N-terminal end of the protein, and the Aux/IAA domain or domain III/IV is usually located at the C-terminus, which is consistent with the findings of previous studies [62,70]. NtARF19 and NtARF40 are very short proteins that have only a B3 domain. Again, as with NtLAV5 and NtLAV4, we can speculate that domain loss occurred in these two proteins during evolution. It is interesting that no expression or very low levels of expression have been detected for *NtARF19*, indicating that *NtARF19* may have become a pseudogene, similar to *AtARF23* in Arabidopsis.

The exon-intron organization analysis, motif analysis, and gene expression analysis further supported our phylogenetic analysis. Genes for the more closely related proteins tend to have similar exon-intron arrangements, and the proteins have similar structures. Unique features in the gene and/or protein structures of some *B3* genes are partially due to the extensive expansion of this gene superfamily through gene duplication or gene fragmentation and loss in other genes.

The results of previous chromosome localization studies have indicated that *REM* genes are inclined to cluster in the genome, and this was observed in 11 plant species including maize, rice, sorghum, and Arabidopsis. A similar situation was also found in tobacco (Figure 2), where the largest cluster consists of nine *REM* genes in a 103.6 kb genomic region located on chromosome 15. Furthermore, we also found that many *B3* genes are also clustered on other chromosomes. The large number of *REM* genes that arose from extensive gene duplication may be responsible for the clustering. However, there is at present no practical experimental evidence to explain this observation.

The expression levels of genes in different tissues and developmental stages are closely related to their function. We investigated the expression patterns of tobacco *B3* genes in different tissues, which contributed to an understanding of the potential functions of these genes. In seedlings, we found that many genes are expressed in the stem and root. In the seedling shoot, several genes in the *REM* and *RAV* subfamilies have high expression levels in addition to the *ARF* genes. In seedling roots, in addition to *NtLAV1* and *NtLAV5*, the highly expressed genes are mainly members of the *ARF* and *REM* subfamilies. In the seedling shoot apex, there are fewer genes with high expression levels than in the other two tissues, but the number of genes that are highly expressed is similar between the three tissues. Most of the genes expressed in the shoot apex had high expression levels and are mainly in the *ARF* subfamily. We hypothesize that the high auxin content in the shoot apex activates transcription of these *ARF* genes. It is worth mentioning that the highly expressed genes in different tissues exhibit tissue-specific expression patterns. In various tissues and organs of adult tobacco, constitutively expressed genes are rarely found in analyses of transcriptome data. Many genes are specifically transcribed at certain developmental stages. *NtARF44* showed high expression levels in young flowers; *NtREM3*, *NtREM43*, and *NtARF2* are highly expressed in mature flowers; and *NtREM27*, *NtREM15* and *NtARF3*, *NtARF11*, and *NtARF15* were expressed at high levels in senescent flowers. However, the expression of *NtARF2* and *NtARF25* increased and then decreased during flower development, peaking in mature flowers, indicating that these two genes can regulate the entire floral developmental process. In Arabidopsis, ARFs also are key factors in floral morphogenesis [77,81,82]. All the evidence indicates that these genes are specifically responsive at different stages of flower development in tobacco (Figure 7). At the same time, some genes are highly expressed in two associated developmental stages. *NtREM19* shows high expression levels in mature and senescent flowers, and *NtREM10* is highly expressed in senescent flowers and dry capsules, indicating that these genes play vital roles in the progression between different stages of development (Figure 7). *NtREM4*, *NtREM55*, *NtREM38*, and *NtLAV3* are highly expressed only in the dried capsule, suggesting that these genes are associated with fruit development. The expression patterns of the tobacco *REM* genes are supported by previous investigations into the expression and transcript features of *BoREM1* and other *REM* genes [101,102,103,105,110,111]. *NtARF11*, *NtARF15*, and *NtRAV1* are highly expressed in young leaves; *NtARF43* is the only gene that showed a high level of expression in senescent leaves and a low level of expression in other tissues. *NtARF10* and *NtARF21* are highly expressed during three stages of leaf development. These results show that *ARF* genes are involved in the development and senescence of tobacco leaves, which is consistent with previous reports [76,77,127]. *RAV* gene expression in tobacco leaves may be the same as the *NGATHA* genes that regulate plant leaf development via the auxin signaling pathway in Arabidopsis [41]. We also found that *NtLAV1* and *NtLAV5*, which are closely related phylogenetically, have similar expression patterns, and their expression levels are very low in almost all tissues assayed. The same things were found for *NtLAV3* and *NtLAV4*, which have a close genetic relationship and are mainly expressed in the dry capsule. *NtLAV3* and *NtLAV4* share a close evolutionary relationship with the *VAL* genes that function to repress the seed maturation program and vegetative development [10,22,23,24].

In addition, we investigated the expression of *B3* genes in tobacco axillary buds after topping, which is an important agronomic practice for producing tobacco leaves of good quality. Because the upper and lower parts of the tobacco plant are affected differently by apical dominance, we sampled the axillary buds from the upper and lower parts of the plants. Compared to the control, many tobacco *REM* genes showed significant up-regulated expression after topping, in both upper and lower axillary buds. However, only four *ARF* genes were up-regulated to some extent. Interestingly, all up-regulated *ARF* genes were in the lower axillary buds, and the expression levels of all three *ARF* genes that showed significant changes in expression levels in the upper axillary buds were down-regulated. It is interesting that all the *ARF* genes with significant down-regulating expression were transcriptional repressors, such as *NtARF22*, *NtARF35*, and *NtARF41* in upper axillary buds, *NtARF28*, *NtARF30*, and *NtARF33* in lower axillary buds, whose middle domain is rich in proline (P), serine (S), and threonine (T). The *ARF* genes with significantly up-regulating expression were transcriptional activators, such as *NtARF32* and *NtARF39* in lower axillary buds, whose middle domain is rich in glutamine (Q)-rich middle region (Figure 10). These results further support the model of gene response to auxin in that the *ARF* genes in the upper axillary buds can respond to high auxin concentrations in the control of apical dominance, and the activity of these proteins are repressed when auxin concentrations decrease as a result of removing apical dominance by topping.

The increased expression of many *REM* genes shows that *REM* genes regulate the development of axillary buds. In addition, the expression level of *NtRAV1* in the upper axillary buds was significantly lower, and *NtRAV2* and *NtRAV5* in the lower axillary buds were down-regulated in response to topping treatment, which indicated that the *RAV* family plays a crucial role in axillary bud growth and development. It is worth noting that the sequence homology between *NtRAV2* and *NtRAV5* is >90%, and the gene structures and protein structures, as well as the changes in expression in response to topping are very similar, indicating that these two genes might have redundant functions. In the *LAV* family, the *NtLAV5* gene was found to be down-regulated in the upper axillary buds and up-regulated in the lower axillary buds after topping. We speculate that *NtLAV5* regulates the growth of axillary buds via auxin pathways. The sum of all the evidence presented here shows that *B3* gene family is very important in tobacco growth and development, especially with respect to axillary bud growth and development. Therefore, our systematic analysis of the *B3* gene family in tobacco will be of great value for future research into the molecular mechanisms that control growth and development in tobacco, especially the growth and development of the axillary buds.

## Figures and Tables

**Figure 1 genes-10-00164-f001:**
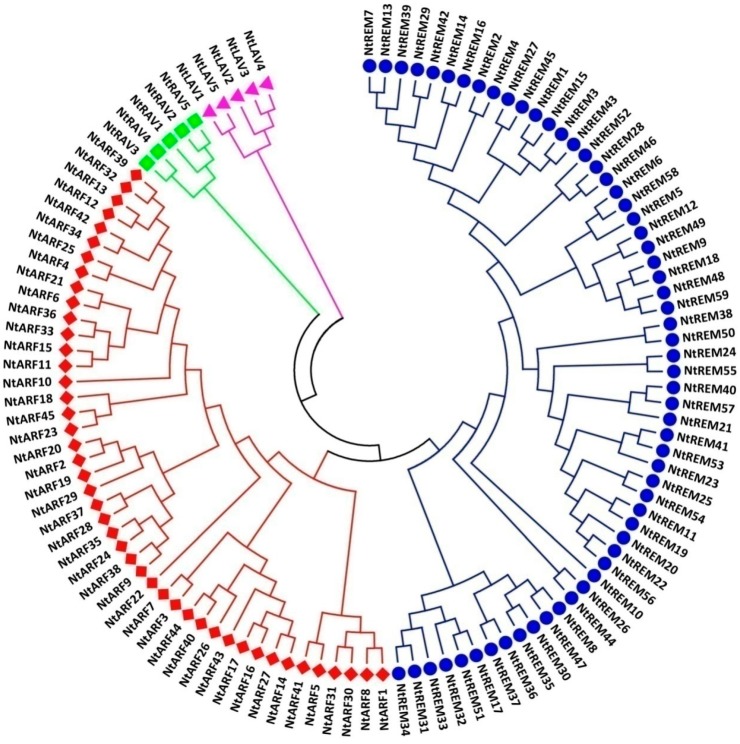
Phylogenetic analysis of B3 proteins in tobacco. The phylogenetic tree was constructed using the ML (Maximum likelihood) method with 1000 bootstrap replications based on the JTT matrix-based model.

**Figure 2 genes-10-00164-f002:**
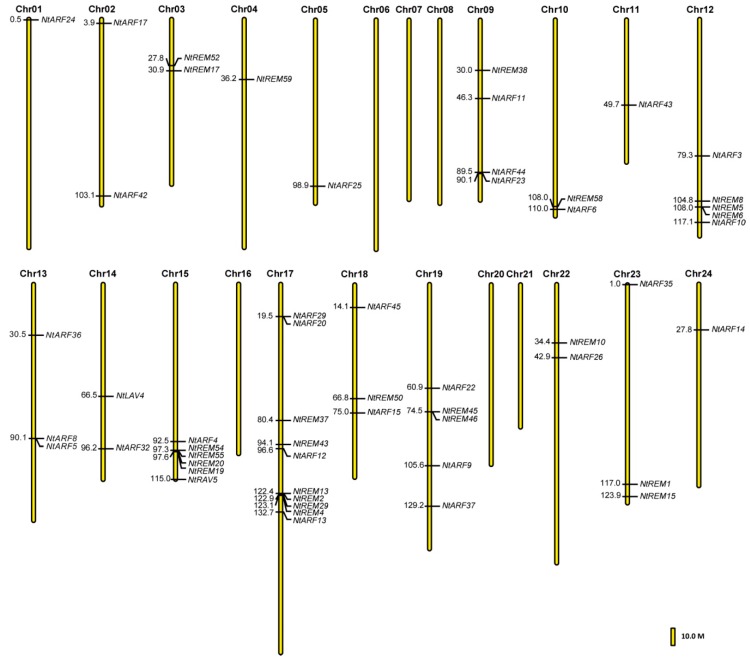
Chromosomal location of tobacco *B3* genes based on their physical positions. Bar = 10.0 Mb.

**Figure 3 genes-10-00164-f003:**
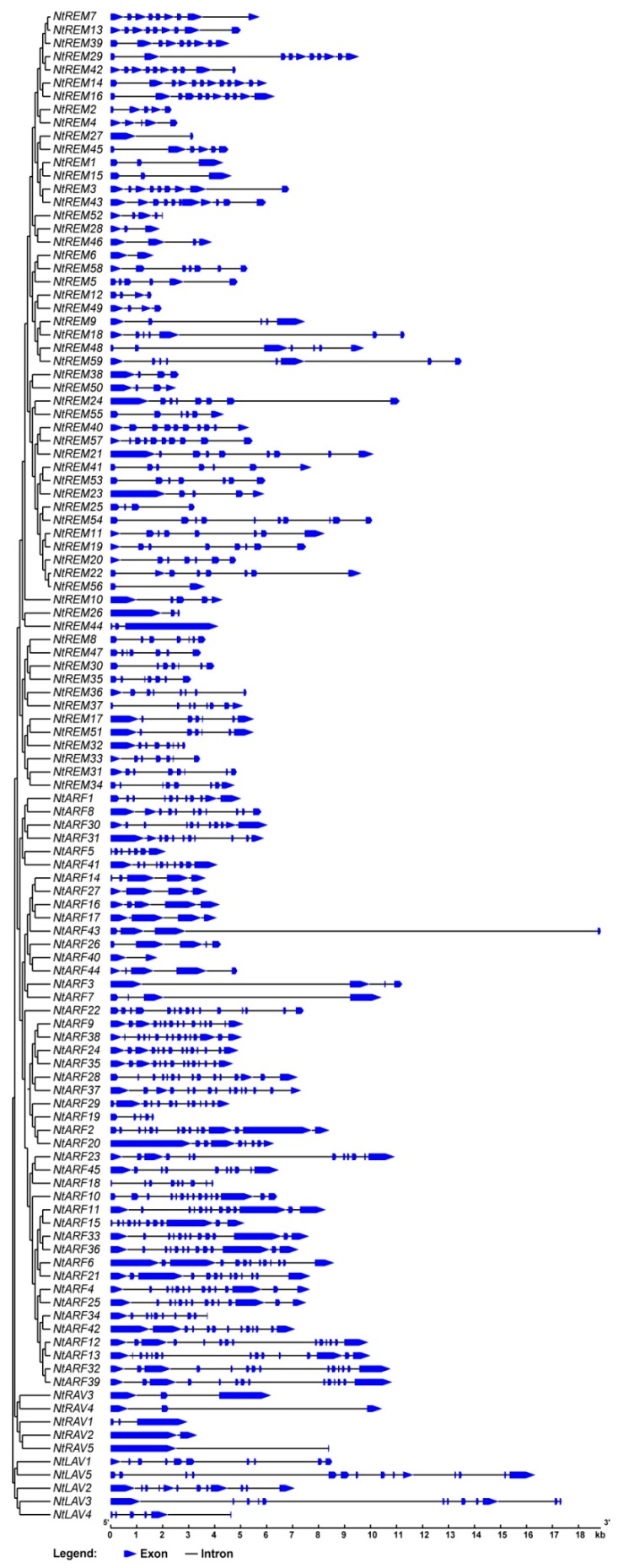
Gene structure analysis of 114 *B3* genes in tobacco according to each family. Exons and introns are represented by colored boxes and black lines, respectively.

**Figure 4 genes-10-00164-f004:**
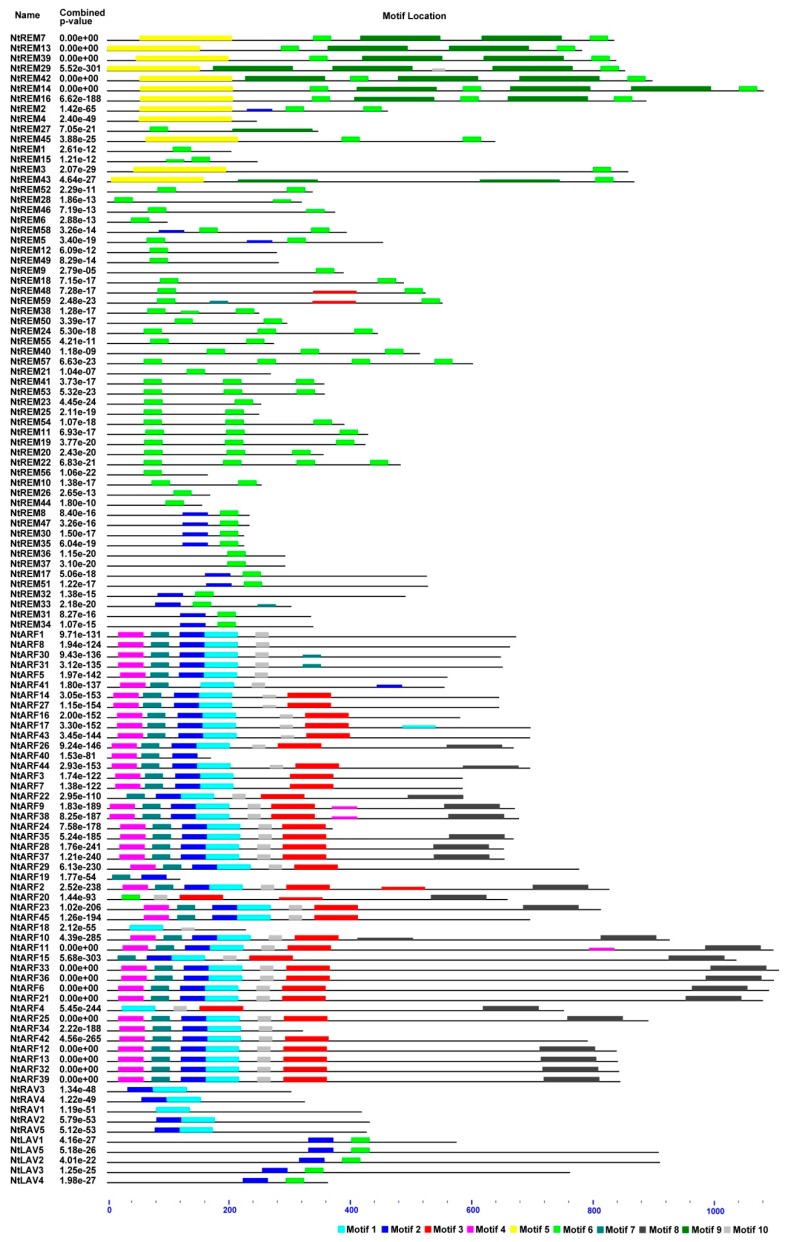
Conserved motifs analysis of B3 proteins according to the phylogenetic relationship. Each motif is represented by a number in a colored box. Box length corresponds to motif length. Motif location was showed.

**Figure 5 genes-10-00164-f005:**
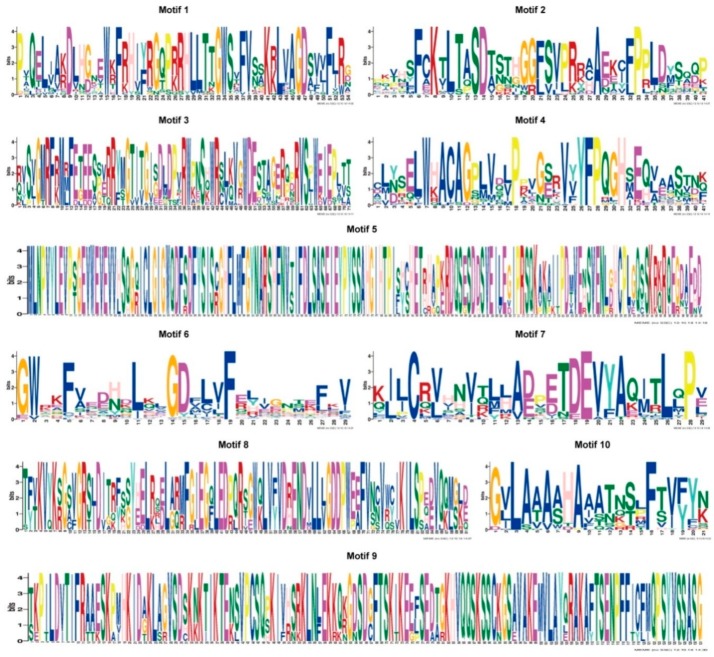
Sequence logos of the ten conserved motifs identified using MEME tools from the B3 proteins in tobacco. Total height of the residues stack indicates the information content of that position in the motif. Height of residues within the stack indicates the probability of each residue at that position.

**Figure 6 genes-10-00164-f006:**
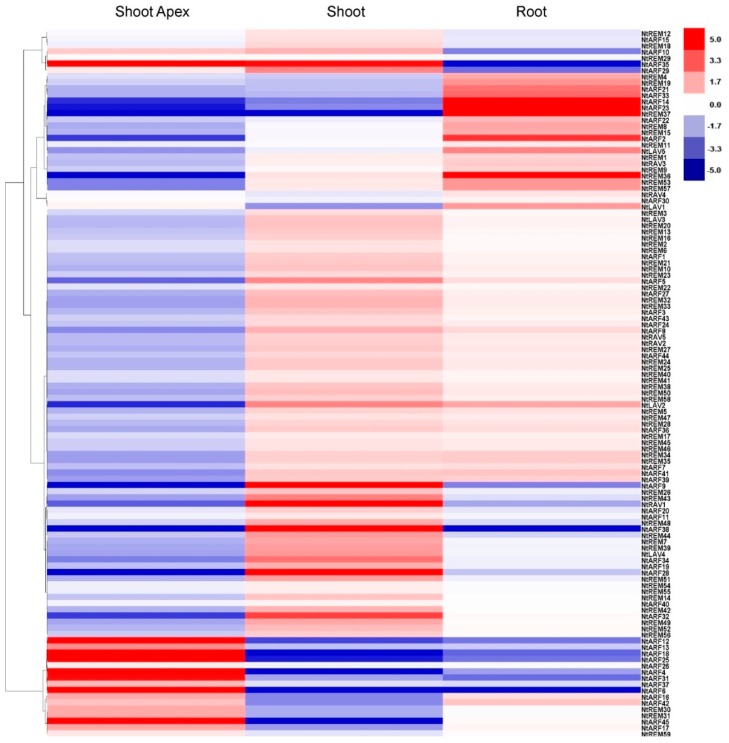
Expression profile analysis of *B3* genes in tobacco tissues from the seedling stage.

**Figure 7 genes-10-00164-f007:**
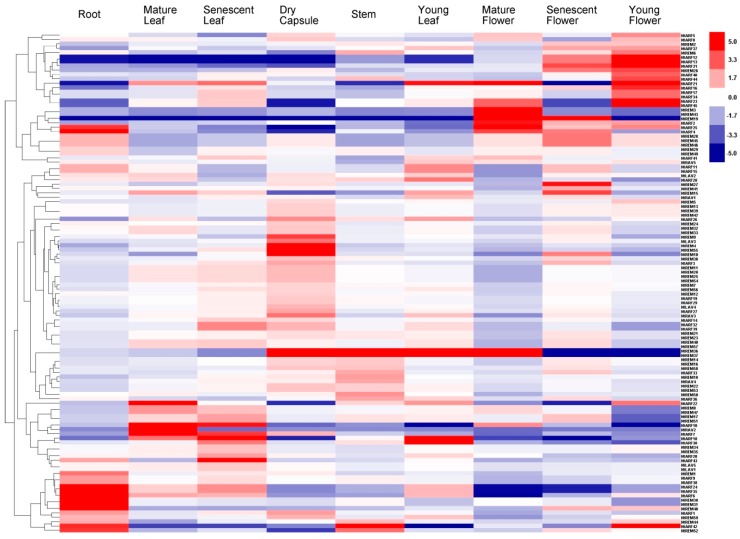
Expression profile analysis of *B3* genes in various tissues from different developmental stages of tobacco.

**Figure 8 genes-10-00164-f008:**
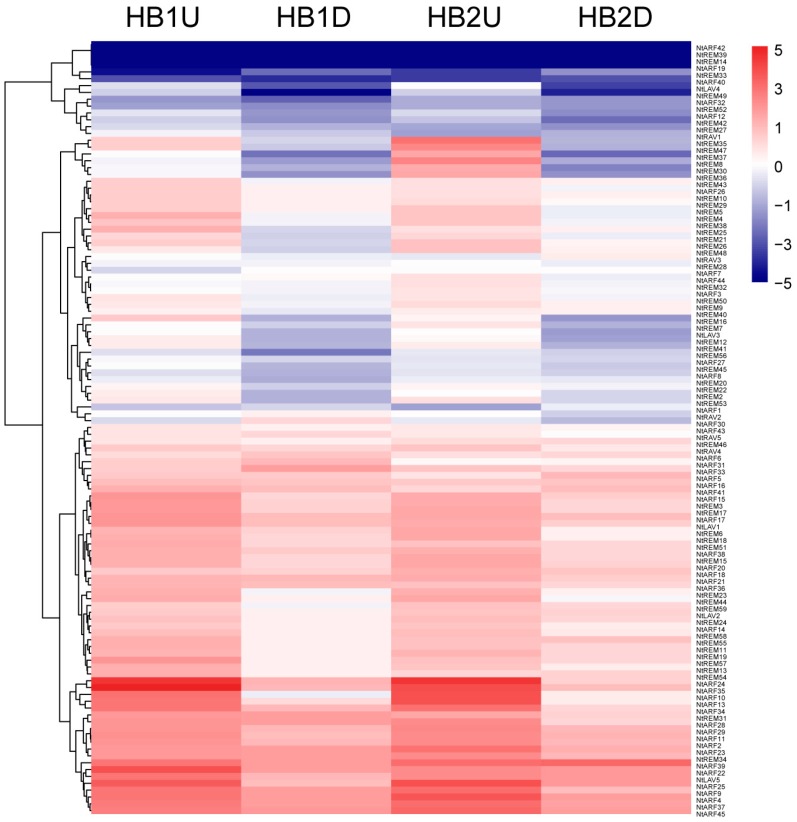
Expression pattern analysis of *B3* genes in tobacco axillary buds in response to topping treatment based on qRT-PCR. HB1U and HB1D indicated the upper and lower axillary buds before topping, respectively. HB2U and HB2D indicated the upper and lower axillary buds after topping, respectively.

**Figure 9 genes-10-00164-f009:**
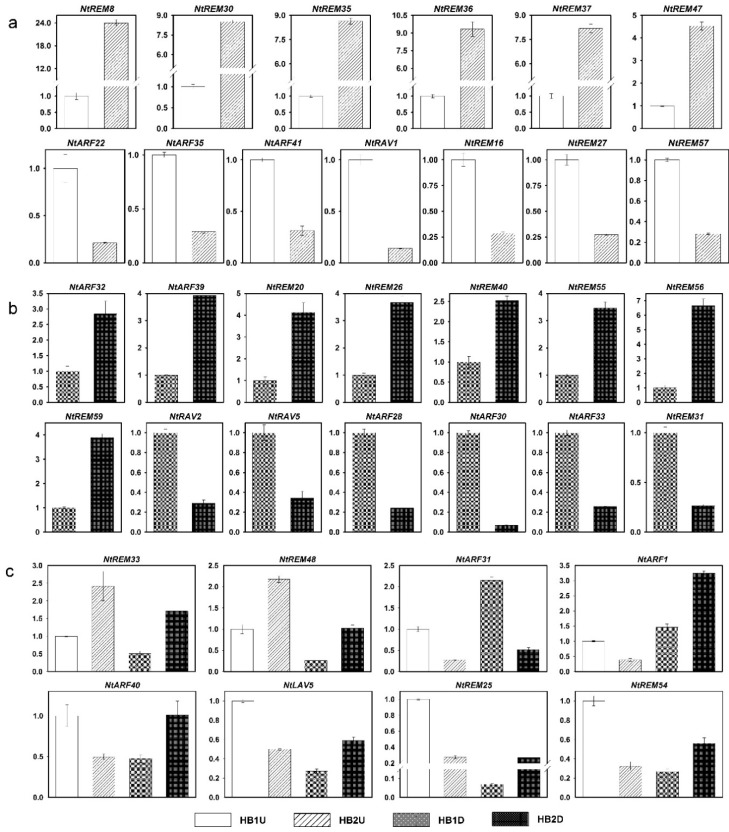
qRT-PCR analysis of the *B3* genes in response to topping with at least 2.5-fold range. (**a**) The expression level of the *B3* genes changed only in upper axillary buds after topping. (**b**) The expression level of the *B3* genes changed only in lower axillary buds after topping. (**c**) The expression level of the *B3* genes changed both in upper and lower axillary buds after topping.

**Figure 10 genes-10-00164-f010:**
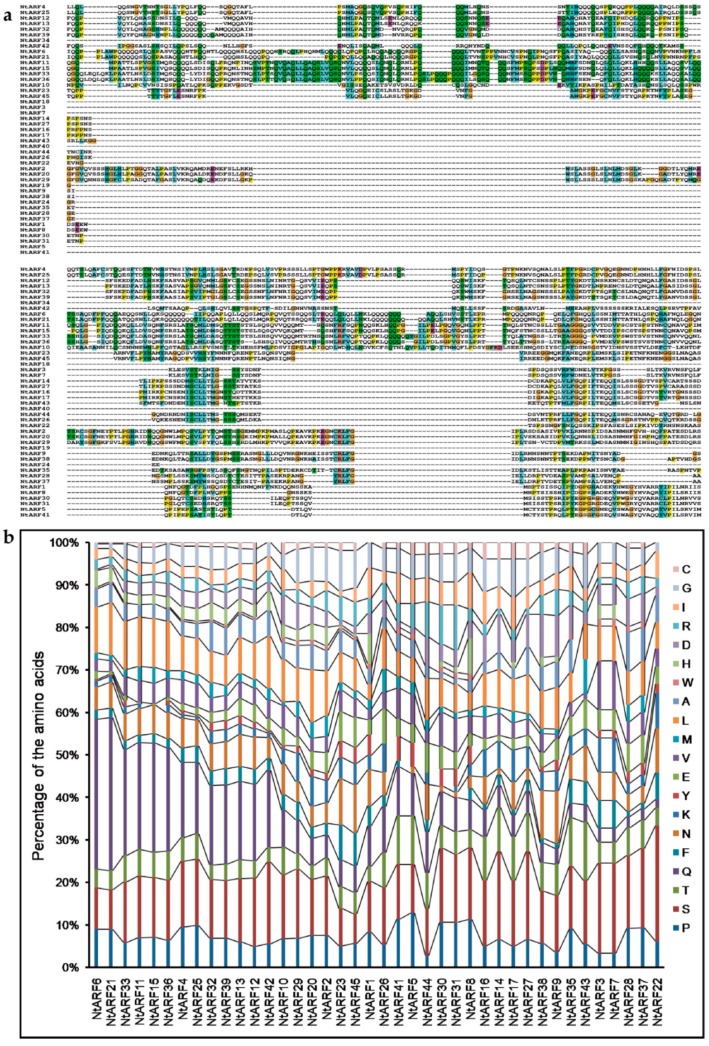
Multiple alignments (**a**) and analysis of amino acid content (**b**) of middle domain in NtARF proteins.

**Table 1 genes-10-00164-t001:** Characteristic features of B3 transcription factor superfamily identified in tobacco.

Protein Name	Locus Name	Gene Length	Exon Number	Amino Acid Length	Molecular Weight	Theoretical pI	WoLF PAORT
NtARF1	mRNA_8768_cds	5017	11	673	76,403.29	5.5	Nuclear
NtARF2	mRNA_123344_cds	8417	15	827	92,057.7	6.34	Nuclear
NtARF3	mRNA_123920_cds	10585	4	585	64,124.39	5.55	Nuclear
NtARF4	mRNA_136897_cds	7668	14	752	84,000.46	6.32	Nuclear
NtARF5	mRNA_10429_cds	2123	7	560	62,854.56	5.99	Nuclear
NtARF6	mRNA_138033_cds	8596	13	1090	122,010.84	6.14	Nuclear
NtARF7	mRNA_143247_cds	10421	4	585	64,156.31	5.64	Nuclear
NtARF8	mRNA_10432_cds	5799	11	663	75,220.24	5.61	Nuclear
NtARF9	mRNA_13156_cds	5103	15	671	75,146.84	5.99	Nuclear
NtARF10	mRNA_20404_cds	6406	15	926	102,414.84	5.42	Nuclear
NtARF11	mRNA_21030_cds	8276	13	1097	121,044.53	6.1	Nuclear
NtARF12	mRNA_828_cds	9910	14	839	93,884.44	5.91	Nuclear
NtARF13	mRNA_28726_cds	9993	15	841	94,136.7	5.96	Nuclear
NtARF14	mRNA_38282_cds	3657	5	645	71,194.66	5.97	Nuclear
NtARF15	mRNA_39086_cds	5147	12	1036	114,794.48	6.12	Nuclear
NtARF16	mRNA_2133_cds	4199	5	581	63,946.73	8.57	Nuclear
NtARF17	mRNA_46303_cds	4080	4	697	76,601.62	7	Nuclear
NtARF18	mRNA_51744_cds	3949	9	228	25,791.46	9.8	Chloroplast
NtARF19	mRNA_56246_cds	1662	5	120	13,615.58	6.28	Cytoplasmic
NtARF20	mRNA_56267_cds	6287	9	659	73,065.38	6.82	Chloroplast
NtARF21	mRNA_58268_cds	7685	13	1080	120,714.47	6.24	Nuclear
NtARF22	mRNA_61599_cds	7427	15	586	66,727.98	9.17	Nuclear
NtARF23	mRNA_63531_cds	10932	12	813	90,516.88	6.01	Nuclear
NtARF24	mRNA_4624_cds	4923	14	371	41,847.43	7.23	Nuclear
NtARF25	mRNA_64415_cds	7527	14	891	98,491.78	6.01	Nuclear
NtARF26	mRNA_67201_cds	4245	5	669	73,902.26	6.94	Nuclear
NtARF27	mRNA_67540_cds	3030	4	645	71,452.8	5.82	Nuclear
NtARF28	mRNA_72472_cds	7201	15	653	72,750.24	6.19	Nuclear
NtARF29	mRNA_74528_cds	4581	13	777	86,319.1	8.21	Nuclear
NtARF30	mRNA_77950_cds	6035	11	648	73,746.25	5.47	Nuclear
NtARF31	mRNA_84970_cds	5899	11	651	74,131.64	5.43	Nuclear
NtARF32	mRNA_86031_cds	10773	14	843	94,349.21	6.15	Nuclear
NtARF33	mRNA_88384_cds	7628	13	1106	122,429.8	5.99	Nuclear
NtARF34	mRNA_91665_cds	3726	10	322	35,652.78	8.7	Chloroplast
NtARF35	mRNA_92678_cds	4707	14	669	74,989.83	6.36	Nuclear
NtARF36	mRNA_93898_cds	7230	13	1098	121,609.98	5.97	Nuclear
NtARF37	mRNA_106346_cds	7331	15	654	72,964.4	6.06	Nuclear
NtARF38	mRNA_108249_cds	5049	15	678	75,608.24	5.98	Nuclear
NtARF39	mRNA_110567_cds	10827	14	845	94,574.25	6.04	Nuclear
NtARF40	mRNA_114883_cds	1793	2	170	18,801.42	5.46	Cytoplasmic
NtARF41	mRNA_115491_cds	4118	10	555	62,458.06	5.48	Nuclear
NtARF42	mRNA_115527_cds	7098	13	791	87,193.17	6.68	Nuclear
NtARF43	mRNA_117690_cds	18861	4	696	77,106.71	6.99	Nuclear
NtARF44	mRNA_119154_cds	4871	5	696	77,152.17	7.25	Nuclear
NtARF45	mRNA_120244_cds	6469	10	696	77,178.19	8.2	Nuclear
NtLAV1	mRNA_123857_cds	8524	10	575	62,891.09	8.38	Nuclear
NtLAV2	mRNA_45860_cds	7079	12	910	100,620.7	6.58	Nuclear
NtLAV3	mRNA_47387_cds	17347	12	762	83,983.75	6.26	Nuclear
NtLAV4	mRNA_91460_cds	4643	6	363	40,056.98	6.16	Nuclear
NtLAV5	mRNA_113848_cds	16337	14	908	99,224.65	6.38	Nuclear
NtRAV1	mRNA_143107_cds	2955	3	419	47,086.53	7.32	Nuclear
NtRAV2	mRNA_64217_cds	3327	2	432	48,228.39	8.21	Nuclear
NtRAV3	mRNA_89347_cds	6164	3	303	34,119.48	7.79	Nuclear
NtRAV4	mRNA_91850_cds	10441	3	325	36,682.32	6.97	Nuclear
NtRAV5	mRNA_121756_cds	8408	2	427	47,396.55	8.21	Nuclear
NtREM1	mRNA_174_cds	4337	3	204	23,183.87	4.5	Nuclear
NtREM2	mRNA_4145_cds	2340	5	462	52,478.62	8.33	Nuclear
NtREM3	mRNA_5127_cds	6868	9	858	101,612.96	9.37	Nuclear
NtREM4	mRNA_6438_cds	2573	5	246	55,365.5	6.09	Nuclear
NtREM5	mRNA_7338_cds	4885	6	454	55,397.2	8.44	Nuclear
NtREM6	mRNA_7342_cds	1659	2	99	32,215.88	9.74	Cytoplasmic
NtREM7	mRNA_12898_cds	5729	9	835	49,752.96	9.17	Nuclear
NtREM8	mRNA_15098_cds	3650	7	234	28,827.84	5.46	Nuclear
NtREM9	mRNA_17560_cds	7480	5	389	23,183.87	4.5	Nuclear
NtREM10	mRNA_27980_cds	3762	5	254	26,819.91	9.56	Cytoplasmic
NtREM11	mRNA_30353_cds	8238	8	429	93,438.97	9.45	Cytoplasmic
NtREM12	mRNA_35402_cds	1570	4	279	11,220.96	8.26	Cytoplasmic
NtREM13	mRNA_37366_cds	5008	9	782	62,632.89	6.84	Nuclear
NtREM14	mRNA_39822_cds	6017	12	1081	43,996.37	9.02	Nuclear
NtREM15	mRNA_41054_cds	4658	3	247	69,342.61	6.66	Nuclear
NtREM16	mRNA_41763_cds	6313	11	888	19,209.82	5.76	Nuclear
NtREM17	mRNA_42571_cds	5521	7	526	31,624.25	9.76	Nuclear
NtREM18	mRNA_46958_cds	11,311	7	488	45,134.83	9.47	Nuclear
NtREM19	mRNA_48978_cds	7520	8	425	41,583.82	8.89	Cytoplasmic
NtREM20	mRNA_49562_cds	4826	7	356	59,256.58	7.59	Cytoplasmic
NtREM21	mRNA_54810_cds	10,122	9	269	33,651.85	9.28	Nuclear
NtREM22	mRNA_54825_cds	9646	8	483	50,571.69	8.89	Cytoplasmic
NtREM23	mRNA_54831_cds	5913	5	253	32,265.84	9.63	Cytoplasmic
NtREM24	mRNA_54839_cds	11,118	8	445	59,248.32	8.1	Nuclear
NtREM25	mRNA_54844_cds	3229	4	250	26,849.94	9.54	Mitochondrial
NtREM26	mRNA_56616_cds	2658	3	169	43,796.4	5.24	Nuclear
NtREM27	mRNA_57793_cds	3180	2	347	72,565.57	8.76	Nuclear
NtREM28	mRNA_57802_cds	1893	3	320	17,770.98	4.93	Nuclear
NtREM29	mRNA_68307_cds	9559	10	853	99,049.56	8.89	Nuclear
NtREM30	mRNA_68657_cds	3991	7	225	41,069.05	8.82	Nuclear
NtREM31	mRNA_68659_cds	4851	8	335	59,213.86	6.36	Nuclear
NtREM32	mRNA_68660_cds	2871	8	491	28,195.38	4.96	Chloroplast
NtREM33	mRNA_68662_cds	3438	7	303	94,586.78	9.44	Nuclear
NtREM34	mRNA_70050_cds	4777	8	339	28,277.8	9.48	Nuclear
NtREM35	mRNA_70052_cds	3095	7	225	33,265.54	9.38	Nuclear
NtREM36	mRNA_71850_cds	5227	8	293	33,240.55	9.5	Nuclear
NtREM37	mRNA_75299_cds	5098	8	293	26,167.04	9.13	Nuclear
NtREM38	mRNA_80332_cds	2619	4	250	38,446.93	9.44	Cytoplasmic
NtREM39	mRNA_82188_cds	4574	9	838	34,953.11	9.33	Nuclear
NtREM40	mRNA_84203_cds	5328	10	515	38,349.91	9.56	Cytoplasmic
NtREM41	mRNA_84209_cds	7730	7	357	26,187.97	8.99	Cytoplasmic
NtREM42	mRNA_87132_cds	4813	10	898	97,709.52	8.82	Nuclear
NtREM43	mRNA_88675_cds	5421	11	868	95,633.22	9.04	Nuclear
NtREM44	mRNA_88789_cds	4139	3	156	37,223.68	4.99	Nuclear
NtREM45	mRNA_91785_cds	4529	6	639	39,751.05	6.87	Nuclear
NtREM46	mRNA_91786_cds	3902	4	375	19,071.44	4.6	Nuclear
NtREM47	mRNA_97289_cds	3476	7	234	29,559.15	9.52	Nuclear
NtREM48	mRNA_98542_cds	9751	7	524	50,779.57	8.2	Nuclear
NtREM49	mRNA_99366_cds	1958	4	282	29,188.64	9.2	Cytoplasmic
NtREM50	mRNA_107285_cds	2517	4	296	31,029.21	9.14	Chloroplast
NtREM51	mRNA_119576_cds	5507	7	528	41,119.08	8.88	Nuclear
NtREM52	mRNA_124977_cds	2000	5	338	52,478.62	8.33	Nuclear
NtREM53	mRNA_136475_cds	5957	7	358	49,339.11	8.34	Cytoplasmic
NtREM54	mRNA_136480_cds	10,066	10	390	54,683.67	7.31	Cytoplasmic
NtREM55	mRNA_136483_cds	4361	6	274	59,060.4	8.38	Cytoplasmic
NtREM56	mRNA_136485_cds	3637	2	165	100,682.42	9.17	Cytoplasmic
NtREM57	mRNA_136491_cds	5465	10	602	27,678.15	4.61	Cytoplasmic
NtREM58	mRNA_140712_cds	5265	7	394	122,461.5	9.28	Nuclear
NtREM59	mRNA_140992_cds	13,507	8	552	88,099.21	9.45	Nuclear

**Table 2 genes-10-00164-t002:** Summary and classification of the *B3* superfamily members identified in the genome of Arabidopsis, rice, tomato, and tobacco.

Species	*ARF*	*LAV*	*RAV*	*REM*	Total
Arabidopsis (*A. thaliana*)	23	6	13	76	118
Rice (*Oryza sativa*)	28	7	16	40	91
Tomato (*Solanum lycopersicum*)	22	4	9	62	97
Tobacco (*N. tabacum*)	45	5	5	59	114
Tobacco (*Nicotiana sylvestris*)	23	4	4	44	75
Tobacco (*Nicotiana tomentosiformis*)	23	4	4	15	46

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
