# Peer review of "Insight into the B3Transcription Factor Superfamily and Expression Profiling of B3 Genes in Axillary Buds after Topping in Tobacco (Nicotiana tabacum L.)"

_genes, 2019, doi:10.3390/genes10020164_

Round 1

Reviewer 1 Report

Dear Authors,

I appreciate the opportunity to review this interesting report on insight into B3 transcription factor superfamily and expression profiling of B3 genes in axillary buds after topping in tobacco. I commend the authors for a number of strengths of their work. The presentation of the paper is clear. The conclusions are supported by the results. However, the 1000 bootstrap used for phylogenetic tree is not enough. Generally, the bootstrap should be over 10.

Author Response

Point 1: '......However, the 1000 bootstrap used for phylogenetic tree is not enough. Generally, the bootstrap should be over 106.'

Responce 1: Actually, the 1,000 bootstrap replicates are enough for the normal phylogenetic analysis. The phylogenetic analysis on many gene families was all performed by using MEGA program with boot-strap value of 1000 replicates, such as ARFs in paper [1] and [2], GRAS in paper [3], etc. 

1.     Zhou, X.; Wu, X.; Li, T.; Jia, M.; Liu, X.; Zou, Y.; Liu, Z.; Wen, F. Identification, characterization, and expression analysis of auxin response factor (ARF) gene family in Brachypodium distachyon. Functional & integrative genomics 2018, 10.1007/s10142-018-0622-z, doi:10.1007/s10142-018-0622-z.

2.     Xiao, G.; He, P.; Zhao, P.; Liu, H.; Zhang, L.; Pang, C.; Yu, J. Genome-wide identification of the GhARF gene family reveals that GhARF2 and GhARF18 are involved in cotton fibre cell initiation. J Exp Bot 2018, 69, 4323-4337, doi:10.1093/jxb/ery219.

3.     Zhang, B.; Liu, J.; Yang, Z.E.; Chen, E.Y.; Zhang, C.J.; Zhang, X.Y.; Li, F.G. Genome-wide analysis of GRAS transcription factor gene family in Gossypium hirsutum L. BMC Genomics 2018, 19, 348, doi:10.1186/s12864-018-4722-x.

Reviewer 2 Report

In this manuscript, the authors identified the complete set of B3 transcription factor genes in tobacco based on sequence similarity and classified them phylogenetically in relation to the Arabidopsis B3 genes. Additionally, the authors analyzed the patterns of expression of all the B3 genes based on available RNA-seq data and showed experimentally the expression of B3 genes in axillary buds in response to topping, demonstrating that different B3 genes may display differential patterns of expression in this response. Interestingly, some ARF genes presumably involved in auxin response decreased their expression after topping, consistently to expected reduced auxin levels.

This is an interesting paper that could contribute to a better knowledge of the plant-specific family of transcription factors B3. Despite this, there are several comments on the manuscript that could be addressed by the authors:

1) Identification and classification of B3 TF family members.

This is a critical aspect in this manuscript but in my opinion, is not conveniently explained in the methods section.

- For example, the HMMER software is used for searching HMM profiles among tobacco proteins, but there is no mention about the parameters used.

- Line 154: ‘The proteins obtained from the results… were subjected to structural domain prediction’. What does ‘structural domain prediction’ mean?

- Line 156: ‘The B3 proteins in tobacco and Arabidopsis were compared to identify orthologs’. How these proteins were compared for identification of orthologs?

- Line 164: ‘Multiple sequence alignments of the predicted B3 protein sequences were conducted using ClustalW and/or Muscle’. I have to say that I do not have any preconceived idea about which method (ClustalW or Muscle) should be used. Moreover, I guess that both methods will yield very similar –if not identical- results. However, it does not seem very consistent to use one or another indistinctly. The authors should check that both methods are similar for some proteins before use them indistinctly. Alternatively, just one method should be used.

2) Number of B3 genes in plants. Table 2 summarizes the number of B3 genes in different species. Arabidopsis, rice and tomato have a similar number of B3 genes, which does not seem quite strange. It is more difficult to understand, however, why tomato and tobacco do have a similar number of B3 genes, given that tobacco is an allotetraploid plant. Moreover, it is even stranger when the number ARFs is double in tobacco than in tomato. An explanation for this issue is discussed in the manuscript but, honestly, the alternatives are no so convincing. For example, ‘tobacco genome has suffered large-scale gene loss following the whole-genome duplication event (line 424)’. It is difficult to understand how REM, RAV and LAV subfamilies may be suffered gene loss but the ARF subfamily did not.

To confirm that the numbers of B3 genes, the authors should perform an identification of tomato B3 using identical methodologies as in tobacco. In addition, gene searches based on TBLASTN could identify novel B3 genes in tobacco that escaped from the initial search using HMMER.

Now it is also available the sequence of the supposed parental species of the allotetraploid tobacco. In these species, the number of ARF should be –more or less- half to that of N. tabacum.

Perhaps more interestingly from an agricultural point of view, side-to-side comparisons of tobacco and tomato B3 genes seem appropriate to better define this superfamily.  

The above strategies could contribute to a better characterization of the B3 superfamily and would explain the apparent differences in gene number between tobacco and tomato.

3) Expression of B3 genes. This part of the manuscript is very interesting and provides expression data for the newly identified genes, based on the use of available RNAseq data and in the experimental quantification of the RNA levels in response to decapitation or topping of plants. However, some questions should be addressed:

- I do not understand why expression data in Fig7 are represented as log ratios (between -5 to +5). It is supposed that expression is FPKM, according to the methods section. Do these ratios correspond to a normalization with a different condition/tissue?  

- Fig 8 and 9 are somehow redundant, since both correspond to the same data. One of them could be moved to Supp. Material.

- Although not necessary and perhaps out of the focus, the manuscript would greatly benefit from the identification of ortholog B3 genes in other species (tomato, Arabidopsis) and the comparison of their expression patterns in different tissues/conditions used available RNAseq data.  

- In the same direction, identification of Arabidopsis-tobacco most likely ARF orthologs may help to the discussion of the expression patterns after topping, in particular of those genes involved in auxin signaling.

Some other comments:

4) To my knowledge, the domain referred in this manuscript to as the ‘auxin response factor domain’ is not generally known like this, but rather as the ‘middle domain’ or ‘MD’. Actually, this is the first time I read ‘auxin response factor domain’.

In relation to the MD, this domain may be a transcriptional activator or repressor domain depending on the composition and, thus, the corresponding ARF transcription factors (TFs) act as activators or repressors, respectively. It would be of interest to predict if the different Nicotiana ARFs likely are activators or repressors, and if these predictions correspond to those of their Arabidopsis orthologs.

5) Subcellular locations of TFs in Table 1 do not seem very relevant. Moreover, we could accept that some TFs may localize in the cytoplasm under particular conditions (keeping them in an ‘inactive’ form), but their biological function is expected to occur in the nucleus. A bit more difficult to accept is the localization of TFs in chloroplasts or mitochondria…

With this respect, are predictions of subcellular locations of ortholog B3 proteins identical in different species?

Author Response

Point 1:  Identification and classification of B3 TF family members.

This is a critical aspect in this manuscript but in my opinion, is not conveniently explained in the methods section.

1) For example, the HMMER software is used for searching HMM profiles among tobacco proteins, but there is no mention about the parameters used.

Responce 1-1: The HMMER software was used with default parameters. The sentence in the method section has been corrected as '......The conserved B3 domain was used to search for B3 protein sequences by HMMER software with default parameters......'.

2) Line 154: ‘The proteins obtained from the results… were subjected to structural domain prediction’. What does ‘structural domain prediction’ mean?

Responce 1-2: The word 'structural' in the text was deleted. The sentence in the method section has been corrected as '......B3 domain in the proteins obtained was predicted and identified again, and the proteins with no B3 domains were eliminated......'.

3) Line 156: ‘The B3 proteins in tobacco and Arabidopsis were compared to identify orthologs’. How these proteins were compared for identification of orthologs?

Responce 1-3: The sentence in the method section has been changed into '......The B3 proteins in tobacco and Arabidopsis were compared to identify orthologs by searching a local database that was built using Arabidopsis B3 genes, and the E-value was set at 1e-10.......'.

4) Line 164: ‘Multiple sequence alignments of the predicted B3 protein sequences were conducted using ClustalW and/or Muscle’. I have to say that I do not have any preconceived idea about which method (ClustalW or Muscle) should be used. Moreover, I guess that both methods will yield very similar –if not identical- results. However, it does not seem very consistent to use one or another indistinctly. The authors should check that both methods are similar for some proteins before use them indistinctly. Alternatively, just one method should be used.

Responce 1-4: The sentence in the method section has been corrected as 'Multiple sequence alignments of the predicted total B3 proteins in tobacco were conducted using ClustalW with the default pairwise and multiple alignment parameters. The protein sequences of each of the LAV, RAV, and ARF subfamilies from both tobacco and Arabidopsis were aligned using ClustalW with default parameters. And multiple sequence alignments of the REM subfamilies from both tobacco and Arabidopsis were performed using Muscle with default parameters, as the alignments of the REMs based on ClustalW were failed to conduct because of the complexity of sequences similarity......'.

 Point 2:  Number of B3 genes in plants. Table 2 summarizes the number of B3 genes in different species. Arabidopsis, rice and tomato have a similar number of B3 genes, which does not seem quite strange. It is more difficult to understand, however, why tomato and tobacco do have a similar number of B3 genes, given that tobacco is an allotetraploid plant. Moreover, it is even stranger when the number ARFs is double in tobacco than in tomato. An explanation for this issue is discussed in the manuscript but, honestly, the alternatives are no so convincing. For example, ‘tobacco genome has suffered large-scale gene loss following the whole-genome duplication event (line 424)’. It is difficult to understand how REM, RAV and LAV subfamilies may be suffered gene loss but the ARF subfamily did not.

To confirm that the numbers of B3 genes, the authors should perform an identification of tomato B3 using identical methodologies as in tobacco. In addition, gene searches based on TBLASTN could identify novel B3 genes in tobacco that escaped from the initial search using HMMER. Now it is also available the sequence of the supposed parental species of the allotetraploid tobacco. In these species, the number of ARF should be –more or less- half to that of N. tabacum.

Perhaps more interestingly from an agricultural point of view, side-to-side comparisons of tobacco and tomato B3 genes seem appropriate to better define this superfamily. 

The above strategies could contribute to a better characterization of the B3 superfamily and would explain the apparent differences in gene number between tobacco and tomato.

Responce 2: The identification of tomato B3 using identical methodologies as in tobacco have been performed in the study of paper [4]. And gene searches based on TBLASTN were used to identify novel B3 genes in tobacco before, which has been supplemented in the method section.

The number of ARF in parental species of the allotetraploid tobacco was both twenty-three, nearly half to that of N. tabacum. The numbers of RAV and LAV subfamilies in parental species were all four, compared with five members in the allotetraploid tobacco. The number of REM subfamilies in parental species was forty-four and fifteen, respectively, in which total number is equal with that in the allotetraploid tobacco. The data have been added into Table 2. The related results were further discussed in the discussion section.

The suggestion on comparisons of tobacco and tomato B3 genes seems to be acceptable, however, the functions of all B3 genes in tomato are poor to study, and most of them are merely based on bioinformatic predictions. In order to preliminarily predict the biological function of tobacco B3 genes through bioinformatic analysis, the Arabidopsis B3 genes are more proper than tomato B3 gene.

 Point 3:  Expression of B3 genes. This part of the manuscript is very interesting and provides expression data for the newly identified genes, based on the use of available RNAseq data and in the experimental quantification of the RNA levels in response to decapitation or topping of plants. However, some questions should be addressed:

1) I do not understand why expression data in Fig7 are represented as log ratios (between -5 to +5). It is supposed that expression is FPKM, according to the methods section. Do these ratios correspond to a normalization with a different condition/tissue? 

Responce 3-1: The ratios correspond to a normalization with a different tissue. The FPKM excel is analyzed using Cluster software, and then processed to be a CDT profile, which can be used to generate the Heat Map by Treeview software. The Constrat parameter in the Treeview software can be set as a normalization to be, such as '5', and all the values of FPKM were normalized referring to the ratios between -5 to +5 according to the different tissues.

2) Fig 8 and 9 are somehow redundant, since both correspond to the same data. One of them could be moved to Supp. Material.

Responce 3-2: Figure 8 showed the entire tendency of variation among different genes and different tissues. And the bar graphs in Figure 9 are shown to highlight the name, category, and definite degree of the genes changed among different treatments. So we think both of them shall exist in the main text.

3) Although not necessary and perhaps out of the focus, the manuscript would greatly benefit from the identification of ortholog B3 genes in other species (tomato, Arabidopsis) and the comparison of their expression patterns in different tissues/conditions used available RNAseq data.

Responce 3-3: It is a wonderful suggestion, and I think those content is applicable for another article with more details.

4) In the same direction, identification of Arabidopsis-tobacco most likely ARF orthologs may help to the discussion of the expression patterns after topping, in particular of those genes involved in auxin signaling.

Responce 3-4: A wonderful suggestion, too. And those content is applicable for another article with more details.

Point 4:  To my knowledge, the domain referred in this manuscript to as the ‘auxin response factor domain’ is not generally known like this, but rather as the ‘middle domain’ or ‘MD’. Actually, this is the first time I read ‘auxin response factor domain’.

In relation to the MD, this domain may be a transcriptional activator or repressor domain depending on the composition and, thus, the corresponding ARF transcription factors (TFs) act as activators or repressors, respectively. It would be of interest to predict if the different Nicotiana ARFs likely are activators or repressors, and if these predictions correspond to those of their Arabidopsis orthologs.

Responce 4: The words 'auxin response factor domain' in the result section has been corrected as 'middle domain'. Some results about the NtARF expression level and transcriptional activator/repressor have been discussed in the discussion section.

Point 5:  Subcellular locations of TFs in Table 1 do not seem very relevant. Moreover, we could accept that some TFs may localize in the cytoplasm under particular conditions (keeping them in an ‘inactive’ form), but their biological function is expected to occur in the nucleus. A bit more difficult to accept is the localization of TFs in chloroplasts or mitochondria…

With this respect, are predictions of subcellular locations of ortholog B3 proteins identical in different species?

Responce 5: The subcellular locations of TFs have been predicted again using the Protein Subcellular Localization Prediction Tool (WoLF PAORT), and the location predictions of tobacco have no changes. When the subcellular locations of TFs are not predicted in the nuclear, the ortholog B3 proteins of Arabidopsis are localized according to the data of TAIR and NCBI. And all the changes on location have been shown in Table S2. The analysis has been added in the result section. The predictions of subcellular locations of ortholog B3 proteins are not completely identical in different species.

4.     Sun, T.; Wang, D.; Gong, D.; Chen, L.; Chen, Y.; Sun, Y. Genome-wide identification and bioinformatic analysis of B3 superfamily in tomato. Journal of Plant Genetic Resources 2015, 16, 806-814, doi:10.13430/j.cnki.jpgr.2015.04.018.

Round 2

Reviewer 2 Report

I still have some comments regarding the response to Point 3.1. In this question, I asked for the units represented in the heat map. The answer ‘The ratios correspond to a normalization with a different tissue’ is still confusing. Which tissue has been considered for normalization? Besides being properly exposed in the Methods section, a reference in the legend of the figure would help the reader ( something similar to ‘the heat map represents log ratios FPKM tissue X/ FPKM tissue reference’, for example). On the other hand, I do not know what ‘CDT profile’ means.

The authors have considered my comments regarding the identification of B3 genes. Actually, they used TBLASTN to search for novel genes. Given that they searched in the mRNA database, they did not find novel B3. I recognize I did not specify in my comments when I suggested using TBLASTN, but this search should be done against the completed genome, instead of mRNAs, to find novel ‘hidden’ B3 genes.